# Salivary Digestion Extends the Range of Sugar-Aversions in the German Cockroach

**DOI:** 10.3390/insects12030263

**Published:** 2021-03-21

**Authors:** Ayako Wada-Katsumata, Coby Schal

**Affiliations:** Department of Entomology and Plant Pathology and W.M. Keck Center for Behavioral Biology, North Carolina State University, Raleigh, NC 27695, USA

**Keywords:** taste, sugar-aversion, German cockroach, glucose, saliva, salivary digestion, feeding behavior, bait, pest control

## Abstract

**Simple Summary:**

The gustatory sense of animals discriminates nutrients from noxious substances. However, enzymes in saliva can digest food components, potentially transforming the quality and taste of foods, and consequently altering feeding behavior. In the omnivorous German cockroach, *Blattella germanica*, feeding is highly stimulated by sugars, which are ubiquitous in its indoor environment. Glucose-averse cockroaches accept complex sugars, but they perceive glucose as a noxious tastant and reject it. We demonstrated that during feeding, saliva digests complex sugars, releasing glucose, and as a result, glucose-averse cockroaches interrupt their feeding on complex sugars. Moreover, we discovered that salivary alpha-glucosidases contributed to the breakdown of complex sugars to glucose. Therefore, the evolutionary adaptation of glucose-aversion, which protects cockroaches from insecticide-containing baits by changing the taste of glucose from sweet to bitter, also dramatically limits the suitability of complex sugars to glucose-averse cockroaches. We highlight the importance of pre-oral and oral saliva-mediated digestion in insect feeding behavior and the importance of considering the effects of saliva in design of sugar-containing insecticide baits.

**Abstract:**

Saliva has diverse functions in feeding behavior of animals. However, the impact of salivary digestion of food on insect gustatory information processing is poorly documented. Glucose-aversion (GA) in the German cockroach, *Blattella germanica*, is a highly adaptive heritable behavioral resistance trait that protects the cockroach from ingesting glucose-containing-insecticide-baits. In this study, we confirmed that GA cockroaches rejected glucose, but they accepted oligosaccharides. However, whereas wild-type cockroaches that accepted glucose also satiated on oligosaccharides, GA cockroaches ceased ingesting the oligosaccharides within seconds, resulting in significantly lower consumption. We hypothesized that saliva might hydrolyze oligosaccharides, releasing glucose and terminating feeding. By mixing artificially collected cockroach saliva with various oligosaccharides, we demonstrated oligosaccharide-aversion in GA cockroaches. Acarbose, an alpha-glucosidase inhibitor, prevented the accumulation of glucose and rescued the phagostimulatory response and ingestion of oligosaccharides. Our results indicate that pre-oral and oral hydrolysis of oligosaccharides by salivary alpha-glucosidases released glucose, which was then processed by the gustatory system of GA cockroaches as a deterrent and caused the rejection of food. We suggest that the genetic mechanism of glucose-aversion support an extended aversion phenotype that includes glucose-containing oligosaccharides. Salivary digestion protects the cockroach from ingesting toxic chemicals and thus could support the rapid evolution of behavioral and physiological resistance in cockroach populations.

## 1. Introduction

Animals engage in food choice, which discriminates between nutrients and noxious substances in their food. The peripheral gustatory system—taste receptor cells in vertebrates and gustatory receptor neurons (GRNs) in insects—convey information from gustatory receptors (GRs) to the central nervous system (CNS) and thus play pivotal roles in food acceptance and rejection. The gustatory system has been well described in humans and model animals, including the mouse and *Drosophila*, in efforts to advance food science, proper growth and wellbeing of pets and livestock, and manipulation of pest populations with insecticides and semiochemicals [1,2,3,4].

The salivary system is an important component in the regulation of animal feeding. In humans, for example, saliva contributes to taste perception and oral health through its involvement in digestion and ingestion of food ingredients, maintenance of mucous membrane integrity, ecological balance of the oral cavity, and direct antibacterial functions [5,6,7]. Insect saliva also figures prominently in physical and chemical functions of a diverse array of feeding strategies [8]. Insects with chewing mouthparts, such as locusts, cockroaches, caterpillars, and beetles, use saliva as a liquid vehicle for digestion and ingestion of solid foods [9]. Saliva components such as apyrase and anophelin of mosquitoes support blood feeding by preventing clotting [10,11]. Salivary proteins of herbivores have been shown to manipulate host plant metabolism to decrease plant secondary-metabolite toxicity and modify tissue structures [12,13]. However, despite the importance of saliva in insect feeding, the role of saliva in modulating gustatory perception and feeding behaviors is poorly understood. As in humans, insect saliva commonly contains digestive amylases, which are capable of breaking down starch into simpler sugars such as maltose. In humans, such low-molecular-weight carbohydrates have different taste qualities from high-molecular-weight carbohydrates, so saliva transforms rather benign complex carbohydrates to sweet-tasting simpler sugars that elicit appetitive responses [14,15]. It remains unclear, however, whether by degrading food components, insect saliva also alters the gustatory information that tastants convey to the CNS through GRNs and thus influence feeding behavior.

Several features of the German cockroach (*Blattella germanica*) make it particularly amenable for investigating the impact of salivary digestion on feeding behavior. It is an omnivorous scavenger, and a serious indoor pest. Geometric analysis of its macronutrient preferences indicates that its optimum dietary protein:carbohydrate ratio is approximately 1:3 to 1:2 [16] and they particularly prefer sugary food as a carbohydrate source in both laboratory and field environments [17,18,19]. Therefore, insecticide-baits have been commonly used since the mid-1980s to control cockroach populations. Simple sugars, including glucose and fructose, often as components of corn syrup, are effective phagostimulants, and enhance the efficacy of baits when combined with insecticides. However, within just a few years of the broad adoption of commercial baits, pest control operators, and then researchers, observed that unrelated cockroach populations avoided eating baits in the field [20,21,22,23,24]. Silverman and Bieman [25] identified the underlying cause of control failures as behavioral resistance to baits based on glucose-aversion (GA), and this was later explained as a gustatory polymorphism in the peripheral gustatory sensilla [25,26]. Whereas normal wild-type (WT) cockroaches process glucose as a phagostimulant by sugar-sensitive-GRNs on the mouthparts, GA cockroaches with this heritable gustatory trait perceive glucose as a deterrent because glucose stimulates deterrent-sensitive-GRNs on the mouthparts. This trait is highly adaptive under toxic-bait pressure, and as GA cockroaches avoid eating toxic-baits containing glucose, they express behavioral resistance, replacing the WT population in the same manner as with insecticide resistance. The aversion to glucose debilitates any baits containing glucose. Interestingly, however, there are reports of behavioral avoidance of toxic-baits with other phagostimulatory sugars, including fructose, galactose, lactose, maltose, and sucrose [20,21]. The mechanisms of such broad sugar-aversion in these mono- and disaccharide-averse cockroaches have not been investigated. It is vital to understand whether these represent independent gustatory polymorphisms to various sugars, or if a common polymorphism can explain most of the observed phenotypes. Answers to this question are important to for our understanding of general insect feeding mechanisms and to improve pest management.

In this study, using WT and GA cockroaches, we carried out a comparative analysis of feeding responses to various sugars in an effort to identify a common mechanism that underlies aversion to a broad range of sugars. We discuss how pre-oral and oral digestion by saliva dramatically changes insect food preference and how this phenomenon might impact pest management of sugar-averse cockroaches.

## 2. Materials and Methods

### 2.1. Two Cockroach Strains

The wild-type strain (Orlando Normal) was collected in Florida in 1947. The glucose-averse strain (T164) was collected in Florida in 1989 and was artificially selected with glucose-containing toxic bait to fix the glucose-averse (GA) trait in this population. The two strains were maintained on rodent diet (Purina 5001, PMI Nutrition International, St. Louis, MO, USA) and distilled water at 27 °C and ~40% RH on a 12:12 h L:D cycle. We tested sexually mature virgin females (5-day old) in bioassays because at this stage females forage extensively in support of oocyte maturation [27].

### 2.2. Acceptance-Rejection Assays and Consumption Assays

We developed two independent assays to assess feeding.

Acceptance-rejection assay. This was a qualitative yes–no assay that evaluated the instantaneous responses of cockroaches to tastants. It involved a small amount of test solution placed on the paraglossae—the most proximal mouthpart where sugar responsive gustatory sensilla have been observed [28]—and a rapid assessment of either its acceptance or rejection. In this assay, the test solution stimulated gustatory sensilla on the paraglossae, with only brief interaction with saliva. Either one-day starved or non-starved insects were restrained in a plastic pipette tip with only the head exposed. Before starting the assay, insects were satiated with water, except when they were tested with deterrents. The paraglossae were carefully touched under a stereo-microscope with a 0.3 µL drop of test solution dyed with 1 mM blue food dye (Erioglaucine disodium salt). Acceptance of the taste substance was defined as ingestion within 1 s, and rejection was defined as lack of ingestion within 1 sec. During ingestion of the stimulus solution, the blue dye could be seen through the clypeus and frons, the translucent front-middle area of the head capsule. The cockroaches ingested <0.01 µL of the test solution before the solution was withdrawn to avoid satiation.

Consumption assay. This assay quantitatively measured the amount of test solution ingested by each insect. Non-starved females were individually restrained in a pipette tip as in the feeding assay. A test solution containing 0.5 mM blue food dye in a microcapillary (10 µL, DWK Life Science Kimble, Millville, NJ, USA) was brought in contact with the paraglossae and the volume consumed was recorded. We observed each female until she stopped drinking, and we considered this a single bout of feeding. Therefore, in this assay, the test solution stimulated gustatory sensilla on the paraglossae, and more of the test solution had longer contact with saliva.

### 2.3. Sugar Acceptance and Consumption

To assess the feeding responses of WT and GA females in the acceptance-rejection assays, monosaccharides (d-glucose and d-fructose), disaccharides (d-maltose, d-trehalose, d-sucrose), and a trisaccharide (d-maltotriose) were tested. All chemicals were prepared at 0.01, 1, 10, 100, and 1000 mM with 1 mM blue food dye. Two groups of females were prepared (see [29]): Group A was prepared using non-starved females to assess their acceptance responses of phagostimulant sugars. Each insect was satiated with water before the test, to ensure that their acceptance of phagostimulants was not due to thirst; these females would reject the negative control (water with blue dye) and various deterrents. For each sugar 21 WT and 31 GA females were tested. Group B was prepared to assess their rejection responses of various tastants. Females were starved for one day without food and water to produce hungry and thirsty females that were motivated to accept water and phagostimulants, but would reject deterrents at high concentration. Data were obtained from 20 GA females for glucose. The effective concentration (EC_50_) of each sugar was obtained from dose-response curves. For Group A, dose–response curves and EC_50_ values were obtained for WT and GA females in response to six and five sugars, respectively. Glucose was tested as a deterrent in GA females in Group B.

In the consumption assay, the amount consumed (µL) of six types of sugar solutions with 1 mM blue food dye was measured at 0, 10, 100, and 1000 mM. Each female received a single concentration of a single test solution. For assays using plain water consumption, 26 WT and 13 GA females were used. For sugar consumption, 21–25 females were tested at each concentration of each test solution.

### 2.4. Effects of Saliva on Sugar Degradation and Feeding

To evaluate the effects of saliva mixed with sugar solutions on the feeding response in the acceptance-rejection assay, saliva from 5 day-old WT and GA females was collected. Individual females were briefly anesthetized with carbon dioxide under a microscope and gently squeezed at the side of the thorax to avoid contamination with regurgitates from the gut. Saliva droplets were collected at the mouthparts into a microcapillary (10 µL, Kimble Glass). Collected fresh saliva was immediately used in experiments. Test solutions were prepared by mixing 3 μL of 200 mM maltose, maltotriose, trehalose, sucrose or 600 mM glucose or fructose with 3 μL of either HPLC-grade water or saliva of WT or GA females. The four tri- and disaccharides were presented to females at a final concentration of 100 mM, and the two monosaccharides (glucose and fructose) were presented at 300 mM in a total volume of 6 μL. These concentrations were chosen based on the EC_70_ in the acceptance-rejection assay. Saliva and sugars were incubated for 5 min at 25 °C. We tested 18–22 females in each strain. To evaluate the effect of saliva alone on the feeding response, either non-starved females or females that were starved for one day without food and water were tested with water only and the mixture of saliva and water at a 1:1 ratio (*n* = 20). All test solutions contained blue food dye.

### 2.5. Involvement of Salivary Glucosidases in Sugar Degradation

To evaluate whether salivary enzymes are involved in the hydrolysis of disaccharides and trisaccharides, the contribution of salivary alpha-glucosidases was tested in the feeding acceptance–rejection assay using the glucosidase inhibitor acarbose (CAS 56180-94-0, Sigma Aldrich, St. Louis, MO, USA). We first confirmed that the range of 0–125 mM acarbose in HPLC-grade water did not disrupt feeding acceptance or rejection of cockroaches. Test solutions were prepared as follows: 2 µL of either HPLC-grade water or saliva of GA females was mixed with 1 µL of either 250 µM of acarbose or HPLC-grade water, then the mixture was added to 1 µL of 400 mM of six different sugar solutions. The total volume was 4 µL with the final concentration of sugar being 100 mM. As before, saliva and sugars were incubated for 5 min at 25 °C and all test solutions contained blue food dye. Only 5 day-old GA females (*n* = 20–25) were tested in each assay.

### 2.6. Salivary Protein and Alpha-Glucosidase Activity

Salivary protein was measured by the Bradford method (Coomassie protein assay kit, Thermo Scientific, 23200, Rockford, IL, USA) to compare differences between sexes and strains. We conducted 6–8 replications with both sexes of the two strains. To estimate alpha-glucosidase activity of saliva, a colorimetric assay using p-nitrophenol (alpha-glucosidase activity colorimetric assay kit, Biovision, K690-100, Milpitas, CA, USA) was carried out. Either 1 µL of HPLC-grade water or 250 mM acarbose was added to 1 µL of saliva and incubated for 5 min at 25 °C. We conducted 5 replications with both sexes of the two strains.

### 2.7. Statistical Analysis

Chi-square tests (α = 0.05), t-tests (α = 0.05) and ANOVA and Tukey’s HSD tests (α = 0.05) were used, as appropriate.

## 3. Results

### 3.1. Feeding Acceptance-Rejection Assays and Consumption of Six Sugars

Feeding acceptance and rejection by five-day-old WT and GA females was assayed using six sugars over a concentration series (Figure 1A). WT females accepted sugars with high sensitivity in the following order, from high to low sensitivity: Maltotriose, maltose, sucrose, trehalose, fructose, and glucose. EC_50_ values for each sugar, except glucose, were similar between WT and GA females (Appendix A). In consumption assays, both strains consumed equal amounts of water, and both consumed the other five sugar solutions in a dose-dependent manner. However, GA females consumed significantly less trehalose, sucrose, maltose, and maltotriose than WT females did (Figure 1B). Fructose consumption was similar in both strains. These results indicate that GA females initiated drinking di- and trisaccharides, but then terminated ingesting these sugars—but not fructose—earlier than WT females.

### 3.2. Effects of Salivary Degradation of Oligosaccharides on Feeding Responses in the Acceptance-Rejection Assay

We hypothesized that in GA females, salivation during feeding transformed the initial acceptance of oligosaccharides to rejection and that this was the result of salivary degradation of oligosaccharides to glucose. At the sensory level, the action of salivary enzymes would change the taste quality from activation of sugar-sensitive-GRNs by phagostimulatory oligosaccharides to activation of deterrent-sensitive-GRNs by glucose, resulting in a decline in acceptance. In both strains, the mixtures of water and sugar were accepted, except glucose in GA females (Figure 2). In control experiments, saliva alone of either WT or GA females did not induce rejection (Appendix A). However, whereas WT females accepted oligosaccharides mixed with saliva, GA females rejected oligosaccharide solutions mixed with the saliva of either WT or GA females. Fructose was accepted by both strains in all treatments, including mixed with saliva. These results indicate that saliva mediates the decline in acceptance of oligosaccharides by GA females.

### 3.3. Salivary Alpha-Glucosidases Degrade Oligosaccharides

We evaluated the involvement of salivary alpha-glucosidases in the rejection responses of GA females to sugars using the alpha-glucosidase inhibitor acarbose. Acarbose inhibits maltases, glucoamylases, and sucrases, and prevents the hydrolysis of maltose, maltotriose, and sucrose to the monosaccharide glucose. Acarbose alone, mixed with sugar, did not influence the acceptance of sugars by GA females (Figure 3). As expected, saliva mixed with oligosaccharides decreased their acceptance. However, acarbose mixed with saliva rescued the acceptance of maltose, maltotriose, and sucrose, but not trehalose. The results indicate that saliva contains several types of sugar-degrading enzymes, including alpha-glucosidases and trehalases, and that these enzymes catalyze the release of glucose, which is perceived as a deterrent and induces rejection behavior in GA females.

### 3.4. Alpha-Glucosidase Activity in Saliva

To evaluate whether alpha-glucosidase activity varies between WT and GA females and between males and females, we measured total salivary protein and alpha-glucosidase activity of both sexes in WT and GA cockroaches. WT females had significantly more saliva proteins than WT males. Total saliva protein of GA female was slightly higher than in GA males, but these differences were not significant (Figure 4 Left). WT females had significantly more saliva protein than GA females (Figure 4 Left). Alpha-glucosidase activity was highest in WT females, and slightly higher (but not significantly different) in GA females than in GA males (Figure 4 Right). Since acarbose inhibited the enzyme activity of saliva in both sexes of both strains, the results support the bioassay results shown in Figure 3. Thus, salivary alpha-glucosidases mediate sugar rejection via enzymatic degradation of maltose, maltotriose and sucrose to glucose. Since alpha-glucosidase activity was found in male saliva, it suggests that GA males also may reject maltose, maltotriose, and sucrose with the same mechanisms that we described for GA females.

## 4. Discussion

### 4.1. A New Paradigm: Salivary Enzymes Transform the Valence of Tastants and Alter Insect Feeding Behavior

Saliva of animals has adaptively evolved to participate in mechanisms that regulate feeding behavior [5,6,7,8]. In insects, saliva functions in digestion of food ingredients, detoxification, and manipulation of host physiology consistent with species-specific food choices and habitats [8]. However, the impact of pre-oral and oral salivary digestion in altering the gustatory sense and feeding behavior is poorly documented.

We demonstrated the importance of salivary digestion in feeding behavior using an extreme gustatory polymorphism in the German cockroach. In response to intense selection with glucose-containing insecticide baits, multiple populations of the German cockroach have independently evolved glucose-aversion [25,29].

The acceptance-rejection feeding response assay using six sugar solutions revealed that all the tested sugars elicited feeding responses in a dose-dependent manner in both WT and GA strains, but glucose elicited rejection in GA females. Wada-Katsumata et al. [28,29,30] reported that glucose, fructose, maltose, maltotriose, and sucrose stimulate sugar-sensitive-GRNs in WT cockroaches and thus mediate acceptance of tastants. On the other hand, in GA cockroaches, glucose stimulates deterrent-sensitive GRNs that drive rejection of glucose, while the other four sugars stimulate sugar-sensitive GRNs and thus drive acceptance of these sugars. In this study, the feeding responses of females from the two strains supported our previous findings, namely that deterrent-sensitive-GRNs in the mouthparts acquired a sensitivity to glucose in GA cockroaches, driving glucose-aversion. The molecular details of this taste polymorphism is currently under study.

The sensitivity and acceptance of sugars in WT and GA females was highest to maltotriose, then maltose, sucrose, trehalose, and fructose; WT females accepted glucose, but GA females rejected it. In comparison with previous work using males [28,29], the order of sensitivity of females to maltotriose and fructose was the same as in males in both strains. The sensitivity of females to maltose was slightly higher than in males in both strains (WT strain: 2.63 mM in females and 8.34 mM in males; GA strain: 3.81 mM in females and 6.95 mM in males). On the other hand, GA females rejected glucose at 2.2 mM, showing 20-times greater sensitivity to glucose than GA males (46.4 mM), whereas WT females and males had similar acceptance sensitivity to glucose [28]. Feeding activity and food consumption in females are modulated by their physiological state and oocyte development, while males show stable feeding activity after sexual maturation [27]. Our results suggest that there are no strong differences in the sugar sensitivity of females and males in both strains, but rejection of glucose in GA cockroaches may reflect sexual differences in physiological state that mediate gustatory processing via deterrent-sensitive-GRNs.

On the other hand, we found a clear difference in sugar consumption between the two strains. Both WT and GA females initially accepted maltose, maltotriose, trehalose, and sucrose solutions as shown in the instantaneous acceptance-rejection feeding response assays in Figure 1A. However, GA females stopped drinking these sugar solutions before they were satiated, resulting in the consumption of small amounts, as shown in Figure 1B. Females of both strains equally consumed the monosaccharide fructose, indicating that these responses were somehow related to the presence of glucose monomers. We therefore considered the possibility that sugar stimulation of gustatory sensilla on the mouthparts induced the secretion of saliva [31], and that digestive enzymes in the saliva hydrolyzed di- and trisaccharides to glucose. The salivary enzyme profile of the German cockroach has not been characterized, but our results demonstrated the activity of alpha-glucosidases that transformed otherwise phagostimulatory oligosaccharides in WT to deterrents in GA females, and thus their rejection in bioassays. The rejection of maltose, maltotriose, and sucrose by GA females was eliminated in the presence of acarbose, an inhibitor of alpha-glucosidases. Enzyme activity assays confirmed that the saliva of both strains had alpha-glucosidase activity. These results indicate that salivary enzymes, such as alpha-glucosidases, which catalyze the hydrolysis of terminal 1,4-linked alpha-d-glucose residues successively from the non-reducing ends of the chains with the release of beta-d-glucose, contribute to degradation of di- and trisaccharides, and longer oligosaccharides, into glucose. Salivation during feeding caused the release of glucose from oligosaccharides, resulting in the interrupted and reduced consumption in GA females. Importantly, acarbose did not prevent the decline of feeding response to trehalose mixed with saliva. This indicates that the saliva contains other enzymes and proteins that catalyze the release of glucose. For example, trehalase has a different protein structure and is a distinct enzyme from alpha-glucosidase [32] and acarbose is not an optimal inhibitor of trehalases. Thus, although we demonstrated this phenomenon with alpha-glucosidase and acarbose, other salivary glycosidases are likely involved.

The salivary glands of cockroaches consist of secretory acini located in the ends of branching ducts through which saliva flows into a distensible reservoir [33]. Salivation of *Periplaneta americana* is controlled by serotonergic and dopaminergic neurons. Serotonin stimulates production of the proteinaceous saliva components, whereas dopamine induces fluid secretion and, in extreme cases, production of protein-free saliva [34]. In our study, we collected saliva by slightly squeezing the thorax and considered it as the pooled saliva in the salivary reservoir. We do not know if the components of the pooled saliva are the same as the actual saliva secreted during feeding. Additionally, although the liquid we collected was clear and likely represented only saliva, we cannot exclude the possibility that it contained traces of material regurgitated from the foregut. Future research should also consider whether cockroaches use regurgitation from the foregut for pre-oral digestion of food.

Alpha-glucosidases are widely distributed in microorganisms, plants, mammals, and insects [35], and in cockroach species, they were detected in the salivary gland of *Nauphoeta cinerea* [36]. Molecular analysis revealed that alpha-glucosidases are found in the hypopharyngeal gland of forager honeybees [37] and the salivary glands of mosquitoes [38] and aphids [39]. It was suggested that insects swallow saliva during feeding and that the secretion of glucosidases from these glands would enable sugar digestion to commence in the crop. Bioinformatic analysis of protein families of the German cockroach by Harrison et al. [40] revealed that there is a large set of alpha-glucosidase genes, suggesting that they may support extreme omnivory with diverse diets in the human environment. We are currently conducting transcriptome and proteome analyses of saliva and salivary glands to characterize the saliva components of the German cockroach and their functions in gustation and feeding behavior. 

### 4.2. Impact of Salivary Enzymes on Sugar-Aversions in Cockroaches

Compared with numerous reports about physiological resistance to insecticides, the mechanisms of behavioral resistance are still poorly documented. The most obvious mechanisms underlying behavioral resistance appear to be the enhanced ability of the olfactory and gustatory systems to detect insecticides, enabling avoidance of active ingredients or minimizing contact with them, thus evading the accumulation of a toxic dose. In insects that are already resistant to a particular insecticide, behavioral responses might not be necessary because physiological resistance mechanisms already provide effective protection [41]. Behavioral resistance has been reported in mosquitoes and houseflies in relation to observations of field and laboratory failures of specific insecticides. In recent cases, the mosquito *Anopheles sinensis*, the primary vector of vivax malaria in China, quickly developed behavioral resistance by avoiding contact with deltamethrin-treated bed nets [42]. Many pyrethroids have spatial repellency, so greater sensitivity to the active ingredient may express phenotypically as behavioral resistance. Imidacloprid baits have rapidly (over only five years) selected for avoidance behavior in the house fly *Musca domestica* in southern California [43,44]. The sensory mechanisms are not clear in these cases, but it was suggested that the insects acquired new or enhanced ability to detect insecticides. Indeed, a common feature in behavioral resistance in many systems is chemosensory adaptations, including polymorphisms, which minimize contact of the resistant population with the active ingredient.

In contrast, glucose-aversion in the German cockroach is driven by a phagostimulant rather than the active ingredient contained in the bait [25,29]. In this study, we showed that salivary digestion of sugars is a pre-ingestion event that processes complex sugars into simple sugars. While GA cockroaches initially accept disaccharides and trisaccharides, they stop ingesting these sugars and therefore consume only small amounts. The conversion of complex sugars to glucose, and the rapid rejection of glucose by GA cockroaches is an efficient mechanism that protects the cockroach from insecticides formulated with sugars as phagostimulants. Previous studies have focused on the effects of glucose on bait performance, and suggested that disparities between the nutritional quality of the bait and food preferences was driven by gustatory processing that in turn impacts the management of GA cockroach infestations [45]. Our findings extend these observations to all baits that contain phagostimulatory oligosaccharides; salivary digestion during feeding would interrupt bait consumption, compromising the efficacy of such baits.

Field-collected cockroaches may express other sugar aversion mechanisms. For example, Wang et al. [20,21] described a strain from Cincinnati Ohio that showed aversions to fructose, glucose, sucrose, and maltose. Because no differences were found in electrophysiological responses of maxillary palps to sucrose and glucose stimulation [46], it was suggested that the changes responsible for GA are encoded in CNS changes [20,21], rather than in peripheral gustatory sensilla, as we proposed [28,29]. Notably, Wang et al. [20,21] also observed behavioral aversion to fructose in the Cincinnati strain, which is not consistent with the model we proposed in this study. However, we recently collected fructose-averse (FA), glucose-averse (GA), and fructose-glucose-averse (FGA) cockroaches and demonstrated that deterrent-sensitive-GRNs in the mouthparts responded to fructose in FA and FGA cockroaches and to glucose in GA and FGA cockroaches. Backcross and introgression studies revealed that FGA is composed of two independent traits, namely GA and FA, and crosses of GA and FA cockroaches produce FGA offspring (unpublished). Therefore, although multiple mechanisms might underlie aversions for various sugars in cockroaches, a parsimonious explanation is that genotypic changes in peripheral sensilla generate aversions to monosaccharides, and salivary digestion of di-, tri-, and oligosaccharides to glucose and fructose extends the GA and FA phenotypes to more complex sugars [28,29].

### 4.3. Sugar Aversions and Cockroach Control

Toxic baits are commonly used to control cockroach populations. By mixing slow-acting insecticides with phagostimulants, better baits are virtually odorless to man, not repellent to cockroaches, and can be readily applied under a wide range of operational conditions [47,48]. However, cockroach interventions with baits are dependent on cockroaches readily ingesting the bait. The presence of heritable aversions to glucose and fructose, coupled with pre-oral and oral salivary digestion of more complex carbohydrates, would greatly extend the range of sugar aversions expressed phenotypically by cockroaches. Because practically all dietary sugars contain glucose, fructose, or both, salivary digestion of general oligosaccharides would likely extend the range of sugar aversions to all sugars and result in lower bait performance in the field. Additional concerns about these sugar-averse cockroaches are that the small amounts of bait consumed due to sugar aversions would result in less active ingredient ingested, which can contribute to faster selection for adaptive physiological resistance mechanisms. Moreover, broad sugar aversions might lessen coprophagy, which is thought to be an important mechanism of secondary kill in cockroaches [49]. Finally, our preliminary results suggest that sugar in sugar-containing baits can serve as a conditioned stimulus, while the odors of sugar-containing baits serve as unconditioned stimuli that allow sugar-averse cockroaches to learn to avoid them through classical conditioning. To mitigate the evolution of physiological and behavioral resistance mechanisms toward baits, it is important to understand the processes that drive feeding, foraging, and reproductive behavior of the German cockroach toward improving the performance of baits. Phagostimulants that are not based on glucose or fructose units would be one promising option. Another option might be to incorporate into baits inhibitors of salivary enzymes that would prevent the degradation of phagostimulatory sugars to glucose and fructose.

## Figures and Tables

**Figure 1 insects-12-00263-f001:**
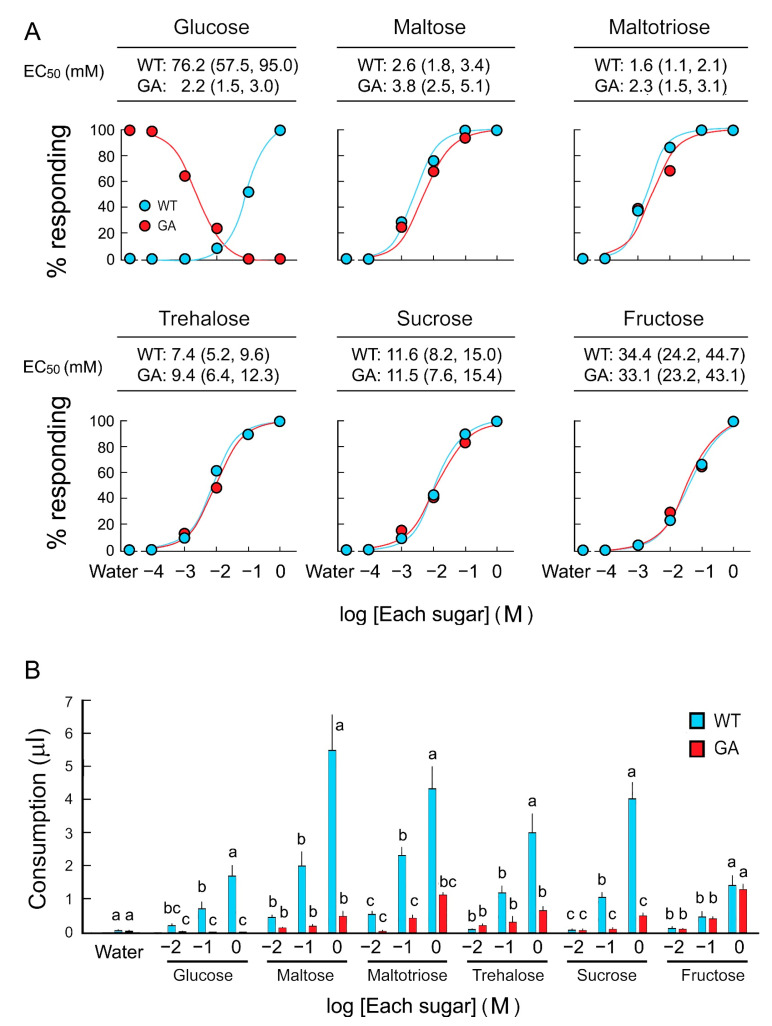
Two feeding assays of *B. germanica* females in response to six sugars. (**A**) Dose-feeding response curves of five-day-old females in the acceptance-rejection assay to assess the effective concentration (EC_50_) of each sugar that stimulates acceptance or rejection. (**B**) Sugar consumption by five-day-old females in a single bout of feeding in the consumption assay. Females ingested sugars in a dose-response manner (Mean ± SE). As expected, glucose-averse (GA) females rejected glucose at all concentrations, but they also consumed significantly lower amounts of di- and trisaccharides than wild-type (WT) females. No significant differences in fructose consumption between WT and GA females across all concentrations indicate that fructose does not interfere with feeding in GA females. Different letters above bars indicate significant differences between treatments within each sugar (ANOVA, Tukey’s HSD, *p* < 0.05) (Appendix A).

**Figure 2 insects-12-00263-f002:**
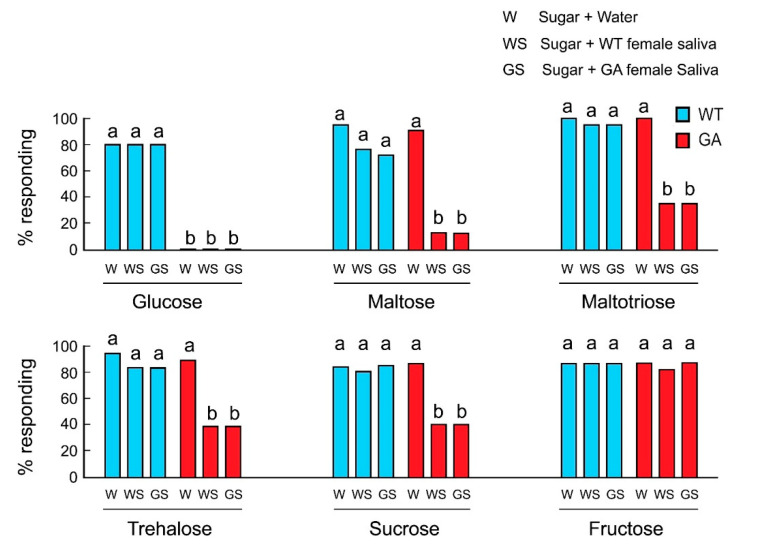
Effects of saliva on feeding responses of *B. germanica* females in acceptance-rejection assays. GA females rejected glucose in all treatments (Mean ± SE). Different letters above bars indicate significant differences among treatments within each sugar (ANOVA, Tukey’s HSD, *p* < 0.05).

**Figure 3 insects-12-00263-f003:**
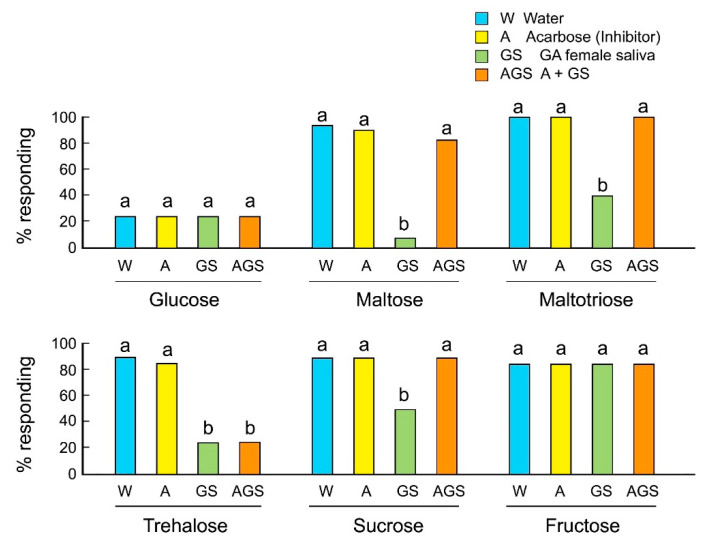
Effects of the alpha-glucosidase inhibitor acarbose on the feeding responses of *B. germanica* females (Mean ± SE). Different letters above bars indicate significant differences within each sugar (ANOVA, Tukey’s HSD, *p* < 0.05) (Appendix A).

**Figure 4 insects-12-00263-f004:**
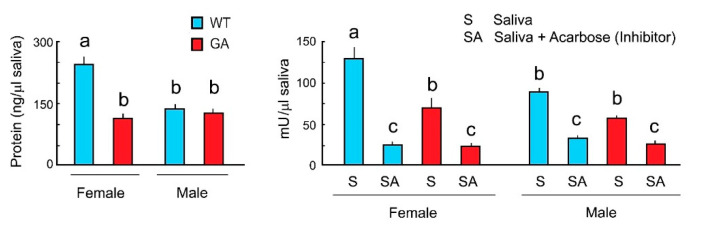
Total salivary protein and alpha-glucosidase activity in both sexes of WT and GA females of *B. germanica*. One unit (U) of alpha-glucosidase is the amount of enzyme that generates 1.0 µmol of p-nitrophenol per min at pH 7.4 at 25 °C. Different letters above bars indicate significant differences (ANOVA, Tukey’s HSD, *p* < 0.05) (Appendix A).

## Data Availability

Data associated with this manuscript have been archived in Dryad Digital Repository (https://doi.org/10.5061/dryad.jsxksn08r).

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
