# Peer review of "Salivary Digestion Extends the Range of Sugar-Aversions in the German Cockroach"

_insects, 2021, doi:10.3390/insects12030263_

Round 1

Reviewer 1 Report

Comments for Authors

Despite the importance of salivary system in insect, it is still unknown the impact of salivary digestion onto the gustatory sense and feeding behaviors. In the manuscript, using wild-type and glucose-aversive strains of German cockroaches, Wada-Katsumata and Schal clearly revealed the function of saliva secretion during early process of feeding behaviors. Because German cockroaches are common pest in the city environment, results of sophisticated behavioral experiments must help to develop controlling methods of the pest insect. Experimental procedures are well organized and obtained results are very clear and beautiful. I have some issues in descriptions of Discussion. This manuscript must be interesting for general readers of Insects and I would like to see the article in Insect after minor revisions.

Major comments:

The description of Discussion is wordy and there are many considerations based on the unpublished results and preliminary experiments. If authors cannot exhibit these data as results in this manuscript, descriptions based on unpublished results must be omitted from the Discussion. In addition, neural mechanisms of gustatory reception in GA-cockroaches repetitively described in Introduction, Results and Discussion. I strongly recommend to shorten the Discussion by omitting the repetitive and unsuitable descriptions.

Minor comments:

Throughout the MS;

Please unify the terminologies sweet and bitter gustatory receptor neurons; for example, “deterrent-sensitive (bitter) GRNs (L89)”, “deterrent-sensitive-GRNs (L243)”, “deterrent-responsive (bitter)-GRNs (L318)” and so on.

L155;

“at each adult day” > “at each adult” or “at each sugar” ?

L186;

Add drug information of acarbose after the sentence.

L212-214;

Add “(SI Table 1)” after the sentence.

L214-215;

The author mentions “No significant differences …” in the sentence, but there are no statistical analysis.

Figure 1;

  • Add (A) and (B) in Figure 1.
  • For easy understanding, it is better to denote EC50 in each panel of Fig. 1A.

All Figure Legends;

Descriptions in Figure legends have been mentioned in main text. Duplicated descriptions and considerations from the obtained results (Ex. The sentence “These results indicate that … (L232-233)” and so on) should be omitted from the Figure Legends.

L245 and 246;

Add “(SI Table 2)” after the sentence. Please appropriately cite Figures, Supplementary Information and results of statistical analysis throughout main text.

L261;

Change “3-3..” to “3-3.”.

L284-285;

Add “(Fig. 4 Left)” after the sentence.

L287-289;

I am very confusing the two successive sentences “Alpha-glucosidase activity was highest in WT females, and slightly higher in GA females than in GA males.” and “We did not find sexual- and strain-differences in enzyme activity.” It needs a clear explanation of the value “mU/ng protein”.

L289-291;

Add the results of statistical analysis after the sentence “We did not find sexual- and strain-differences in enzyme activity.”.

L299;

What is the word “ac Table 0”?

L366;

What is the word “(EC 3.2.1.3)”?

L375-376;

The author mentioned that glucosidases are produced in midgut in the American cockroach. In the American cockroach, saliva produced by salivary gland and stored in salivary reservoir, and then secreted into mouse parts via salivary duct. Therefore, in the American cockroach, the salivary system looks completely independent from the midgut system. The authors obtained German cockroach saliva by squeezing the thorax. Therefore, I cannot deny the possibility that the glucosidase detected in this study is derived from the midgut digestives but not saliva. Throughout the MS, there are a few descriptions of the cockroach salivary systems. Cellular, neural and pharmacological mechanisms of salivary secretion have been well studied in the other species of cockroach Periplaneta americana (See articles from Dr. Walz and Dr. Blenau groups). 

Author Response

Major comments:

The description of Discussion is wordy and there are many considerations based on the unpublished results and preliminary experiments. If authors cannot exhibit these data as results in this manuscript, descriptions based on unpublished results must be omitted from the Discussion. In addition, neural mechanisms of gustatory reception in GA-cockroaches repetitively described in Introduction, Results and Discussion. I strongly recommend to shorten the Discussion by omitting the repetitive and unsuitable descriptions.

Response: We re-organized the Discussion by omitting the repetitive descriptions, as suggested. Because this paper is part of a special issue, we feel that it is appropriate to mention what other research is being conducted on this system, including on fructose-averse cockroaches. Reference to unpublished work does not appear to conflict with journal policy.

Minor comments:

Throughout the MS; Please unify the terminologies sweet and bitter gustatory receptor neurons; for example, “deterrent-sensitive (bitter) GRNs (L89)”, “deterrent-sensitive-GRNs (L243)”, “deterrent-responsive (bitter)-GRNs (L318)” and so on.

Response: Good point. Thank you. We unified the terms “deterrent-sensitive-GRNs” and “sugar-sensitive-GRNs”.

L155; “at each adult day” > “at each adult” or “at each sugar” ?

Response: We changed it to “of each sugar”

L186; Add drug information of acarbose after the sentence.

Response: We added the CAS# and supplier.

L212-214; Add “(SI Table 1)” after the sentence.

Response: Added (SI Table 1).

L214-215; The author mentions “No significant differences …” in the sentence, but there are no statistical analysis.

Response: We modified the sentence to “EC50 values for each sugar, except glucose, were similar between WT and GA females (SI Table 1).”

Figure 1; Add (A) and (B) in Figure 1.

For easy understanding, it is better to denote EC50 in each panel of Fig. 1A.

Response: We modified Figure 1, as suggested.

All Figure Legends; Descriptions in Figure legends have been mentioned in main text. Duplicated descriptions and considerations from the obtained results (Ex. The sentence “These results indicate that … (L232-233)” and so on) should be omitted from the Figure Legends.

Response: We eliminated the duplicated description from all Figure legends.

L245 and 246; Add “(SI Table 2)” after the sentence. Please appropriately cite Figures, Supplementary Information and results of statistical analysis throughout main text.

Response: Added (SI Table 2).

L261; Change “3-3..” to “3-3.”.

Response: Changed

L284-285; Add “(Fig. 4 Left)” after the sentence.

Response: Added

L287-289; I am very confusing the two successive sentences “Alpha-glucosidase activity was highest in WT females, and slightly higher in GA females than in GA males.” and “We did not find sexual- and strain-differences in enzyme activity.” It needs a clear explanation of the value “mU/ng protein”.

Response: We removed the sentence “We did not find sexual- and strain-differences in enzyme activity.” to avoid confusion. The definition of U was added to the figure legend.

L289-291; Add the results of statistical analysis after the sentence “We did not find sexual- and strain-differences in enzyme activity.”.

Response: We removed the sentence “We did not find sexual- and strain-differences in enzyme activity.” to avoid confusion.

L299; What is the word “ac Table 0”?

Response: This was a typio. We modified the sentence.

L366; What is the word “(EC 3.2.1.3)”?

Response: It is IUBMB Enzyme Nomenclature. We removed it to avoid confusion.

L375-376; The author mentioned that glucosidases are produced in midgut in the American cockroach. In the American cockroach, saliva produced by salivary gland and stored in salivary reservoir, and then secreted into mouse parts via salivary duct. Therefore, in the American cockroach, the salivary system looks completely independent from the midgut system. The authors obtained German cockroach saliva by squeezing the thorax. Therefore, I cannot deny the possibility that the glucosidase detected in this study is derived from the midgut digestives but not saliva. Throughout the MS, there are a few descriptions of the cockroach salivary systems. Cellular, neural and pharmacological mechanisms of salivary secretion have been well studied in the other species of cockroach Periplaneta americana (See articles from Dr. Walz and Dr. Blenau groups).

Response: We could not eliminate the possibility that the glucosidase detected in this study was derived from digestive sources in addition to saliva. In the Discussion, we clarified the technical concerns of collecting saliva and added new references. Additionally, we mentioned the possibility that cockroach saliva contains alpha-glucosidase using other references. 

Reviewer 2 Report

This study demonstrated that glucose released from oligosaccharides by enzymatic hydrolyses with salivary alpha-glucosidase induces a deterrent behavior of feeding to GA cockroaches (Blattella germanica), while it acts as phagostimulant to wild-type cockroaches. The result explains the common background of an aversion to multiple sugar containing glucose residues and is expected to contribute to improvement of insecticide-baits for cockroaches. This is an interesting paper, which was written concisely. The experiments were well-designed and conducted appropriately. I have just a few minor comments on this paper.

  1. Although it is reasonable to assume that the salivary alpha-glucosidase play a role in feeding responses, the saliva contains other enzymes and proteins that might cause unknown effect with glucose (as no complete rejection was observed other than glucose alone). In this context, I wondered why the authors used commercially available alpha-glucosidase to confirm the result.
  2. Acarbose mixed with saliva did not rescue the acceptance of trehalose, but this reason was not discussed. Alpha-glucosidases (EC 3.2.1.3 or EC 3.2.1.20) are distinct enzymes from trehalases (EC 3.2.1.28, which sometimes belong to different CAZy families from alpha-glucosidases, suggesting different protein structures; also see Shukla et al. 2015. Glycobiology 25, 357-367), and probably acarbose couldn’t be an inhibitor against trahalases. Thus, alpha-glucosidase was not a sole enzyme involved in this phenomenon.
  3. This paper explained mechanisms of glucose- and fructose-aversion, but what about galactose-aversion described in L95? Is there possibility that isomers (such as galactose and mannose) cause the same effect?

Author Response

Responses to the comments of Reviewer 2

  1. Although it is reasonable to assume that the salivary alpha-glucosidase play a role in feeding responses, the saliva contains other enzymes and proteins that might cause unknown effect with glucose (as no complete rejection was observed other than glucose alone). In this context, I wondered why the authors used commercially available alpha-glucosidase to confirm the result.

Response: Alpha-glucosidases of animals are commonly known as the major enzymes that hydrolyze terminal non-reducing (1→4)-linked alpha-glucose residues to release alpha-glucose molecules. Therefore, to hydrolyze maltose, maltotriose and sucrose, in this study, we focused on determining whether alpha-glucosidase-like enzymes are involved in sugar degradation. We used a commercially available acarbose as an alpha-glucosidase inhibitor in bioassays and enzyme activity analysis, but we did not use a commercially available alpha-glucosidase except for making the calibration curves using an alpha-glucosidase activity colorimetric assay kit (Biovision). We agree with the comments “the saliva contains other enzymes and proteins that might cause unknown effects with glucose” and this is also suggested by the effects of trehalose. Further details on salivary enzymes will be revealed through proteome and transcriptome analysis in future studies. What is clear in our work is that saliva degrades oligosaccharides and this reaction is inhibited, at least in part, by acarbose, implicating alpha-glucosidases. Other salivary proteins likely are also involved.

  1. Acarbose mixed with saliva did not rescue the acceptance of trehalose, but this reason was not discussed. Alpha-glucosidases (EC 3.2.1.3 or EC 3.2.1.20) are distinct enzymes from trehalases (EC 3.2.1.28, which sometimes belong to different CAZy families from alpha-glucosidases, suggesting different protein structures; also see Shukla et al. 2015. Glycobiology 25, 357-367), and probably acarbose couldn’t be an inhibitor against trahalases. Thus, alpha-glucosidase was not a sole enzyme involved in this phenomenon.

Response: We modified this section to reflect the idea that the saliva contains other enzymes and proteins.

  1. This paper explained mechanisms of glucose- and fructose-aversion, but what about galactose-aversion described in L95? Is there possibility that isomers (such as galactose and mannose) cause the same effect?

Response: Although we have not tested mannose, we previously confirmed that both WT and GA cockroaches reject galactose (Wada-Katsumata et al., 2013). Therefore, the galactose-rejection trait is not considered atypical for the gustatory system of the cockroach.
